# Heme Regulatory Motif of Heme Oxygenase-2 Is Involved in the Interaction with NADPH–Cytochrome P450 Reductase and Regulates Enzymatic Activity

**DOI:** 10.3390/ijms26052318

**Published:** 2025-03-05

**Authors:** Masakazu Sugishima, Tomoichiro Kusumoto, Hideaki Sato, Hiroshi Sakamoto, Yuichiro Higashimoto, Ken Yamamoto, Junichi Taira

**Affiliations:** 1Department of Medical Biochemistry, Kurume University School of Medicine, 67 Asahi-machi, Kurume 830-0011, Fukuoka, Japan; hsato@med.kurume-u.ac.jp (H.S.); yamamoto_ken@med.kurume-u.ac.jp (K.Y.); 2Department of Bioscience and Bioinformatics, Graduate School of Computer Science and Systems Engineering, Kyushu Institute of Technology, 680-4 Kawazu, Iizuka 820-8502, Fukuoka, Japan; t0kusumo@bio.kyutech.ac.jp (T.K.); sakakan@bio.kyutech.ac.jp (H.S.); 3Department of Chemistry, Kurume University School of Medicine, 67 Asahi-machi, Kurume 830-0011, Fukuoka, Japan; higashiy@med.kurume-u.ac.jp

**Keywords:** allosteric regulation, crosslink, enzymatic analysis, heme regulatory motif, protein–protein interaction

## Abstract

Mammalian heme oxygenase (HO) catalyzes heme degradation using reducing equivalents supplied by NADPH–cytochrome P450 reductase (CPR). The tertiary structure of the catalytic domain of a constitutively expressed isoform of HO, HO-2, resembles that of the inductive isoform, HO-1, whereas HO-2 has two heme regulatory motifs (HRM) at the proximal portion of the C-terminus, where the disulfide linkage reflects cellular redox conditions and the second heme binding site is located. Here, we report the results of crosslinking experiments, which suggest that HRM is located near the FMN-binding domain of the CPR when it is complexed with HO-2. The enzymatic assay and reduction kinetics results suggest that heme-bound HRM negatively regulates HO-2 activity in vitro. Cellular redox conditions and free heme concentrations may regulate HO-2 activity.

## 1. Introduction

Heme is an essential cofactor for all organisms, whereas free heme is a potent prooxidant that produces reactive oxygen species. Heme oxygenase (HO, EC 1.14.14.18), which is located on the endoplasmic reticulum membrane, catalyzes heme degradation to produce biliverdin, carbon monoxide (CO), and ferrous ion [1,2,3]. The major physiological roles of HO in mammalian cells are the recycling of iron to maintain iron homeostasis and defense against oxidative stress through the degradation of excess free heme as a prooxidant and the production of bilirubin as a potent antioxidant. CO functions as a gaseous secondary messenger with anti-inflammatory, anti-apoptotic, and vasodilatory activities [4,5]. Two isoforms of HO occur in mammals: HO-1, an inducible isoform, and HO-2, a constitutive isoform. A variety of endogenous and exogenous stimuli, such as heat, heme, UV irradiation, oxidative stress, hypoxia, and iron starvation induce the expression of HO-1. HO-1 is highly expressed in the tissues involved in the degradation of senescent red blood cells, such as the spleen, liver, and bone marrow. In most other tissues, HO-1 is undetectable under basal conditions but can be rapidly induced by stimuli. In contrast, HO-2 is ubiquitously and constitutively expressed, with the highest expression identified in the brain and testes [6]. Although its cytoprotective function is believed to play a major role in HO-1, the physiological role of HO-2 is not yet fully understood, although an oxygen sensor in the carotid body mediated by the BK_Ca2+_ channel has been proposed [7]. The enzymatic activity of HO-2 was observed to increase in vivo in response to various stimuli without transcriptional modulation [8], suggesting that HO-2 enzymatic activity is likely regulated by a unique mechanism.

The HO reaction proceeds via a multistep mechanism. The first step is the oxidation of heme to α-hydroxyheme, requiring O_2_ and two reducing equivalents. Then α-hydroxyheme is converted to α-verdoheme with the concomitant release of hydroxylated α-meso carbon as CO. The third step is the porphyrin ring cleavage of α-verdoheme to produce biliverdin–iron chelate, for which O_2_ and reducing equivalents are also required. In the last step, the iron in the biliverdin–iron chelate is reduced, and ferrous ion and biliverdin are finally released from HO. The details of the reaction mechanism were clarified based on the tertiary structures of HO-1 in complex with its substrates, reaction intermediates, and product [9,10,11,12,13,14,15,16,17,18]. The tertiary structure of heme-bound HO-2 (heme–HO-2) resembles that of heme-bound HO-1 (heme–HO-1) [19], implying that the reaction mechanism of HO-2 is similar to that of HO-1.

For the HO reaction to proceed, reducing equivalents supplied by NADPH–cytochrome P450 reductase (CPR, EC 1.6.2.4) are indispensable. CPR is a member of a family of diflavin reductases that catalyze electron transfer from NADPH to the heme groups in their redox partners via FAD and FMN bound to CPR [20,21]. CPR is composed of three domains: the FMN-binding domain, the ferredoxin–NADP^+^ oxidoreductase (FNR)-like domain, and the connecting domain. The latter two domains jointly form the FAD-binding domain. The FAD- and FMN-binding domains are connected by a flexible hinge (Gly232-Arg243 in the rat CPR). NADP^+^, FAD, and FMN were in close proximity in the first reported CPR structure, which is considered a closed conformation. The closed conformation is suitable for intramolecular electron transfer [22] and is predominant in the oxidized state [23]; however, it is not favorable for intermolecular electron transfer because the redox partner must be positioned close enough to the FMN to accept a reducing equivalent. Hamdane et al. developed a rat CPR mutant in which four consecutive residues in the hinge region (Thr236-Glu239 in rat CPR) were removed (hereafter referred to as ΔTGEE) [24]. The crystal structure of ΔTGEE demonstrates three remarkably extended conformations (open conformation). The structures of ΔTGEE indicate that the FMN-binding domain is highly mobile compared to the rest of the molecule; thereby, the open conformation of CPR can bind various redox partners: cytochrome P450s and HO.

Our previous study determined the crystal structure of ΔTGEE complexed with heme–rat HO-1 [25] and clarified that the affinity between them is regulated by NADP^+^/NADPH binding [26,27,28]. The results of the NMR analysis of human HO-2 [29] suggested that the interaction between CPR and HO-2 is consistent with the crystallographic result of ΔTGEE and HO-1. However, the details of the interaction between CPR and HO-2 are still unclear because HO-2 has two heme-regulatory motifs (HRM) (Lys263-Ala268 and Asn280-Thr285) (Appendix A), in which the disulfide linkage between these cysteines depends on cellular redox conditions, and the binding of the second heme has been reported [30,31,32,33,34,35]. The thiol/disulfide redox switch of HO-2 has been implicated in its reported role of HO-2 in oxygen sensing via BK_Ca2+_ channels in the carotid body [6]. Heme binding to HRM is commonly observed in heme-regulated transcription factors such as Rev-erbβ and Bach1, an enzyme involved in heme synthesis (ALAS), and a translation regulator of ferritin and transferrin (IRP) [36,37]. The second heme molecule bound to HRM may move to the catalytic site of HO-2 to immediately supply heme as a substrate [32]. NMR analysis of human HO-2 may not reflect this control by HRM, although the interaction between CPR and HO-2 is expected to be similar to that between CPR and HO-1. To clarify this uncertainty, we performed crosslinking experiments using truncated soluble forms of rat CPR (sCPR) and HO-2 (sHO-2). In this study, 28 unstructured residues were truncated from the N-terminus of sHO-2. Hereafter, the N-terminal-truncated sHO-2 is referred to as tsHO-2 (Appendix A). We previously demonstrated that Val146 and Lys177 of sHO-1 are located near Thr88 and Gln517 of sCPR, based on disulfide-bonded complex formation between recombinant proteins with cysteine mutations [25]. Based on the validity of the crosslinking experiments, we confirmed that amino acid residues are involved in complex formation between tsHO-2 and sCPR, with cysteine introduced at positions where disulfide bonds are expected to form if the interaction mode between heme–tsHO-2 and sCPR is similar to that between heme–sHO-1 and sCPR. The present results suggest that the mode of interaction between heme–tsHO-2 and sCPR is partially similar to that between heme–sHO-1 and sCPR and that HRM is also involved in its interaction with sCPR. Furthermore, we biochemically demonstrated that HRM negatively regulates tsHO-2 activity in vitro by controlling its interactions with the sCPR.

## 2. Results

### 2.1. Crosslinking Experiments

Figure 1A,B show the results of crosslinking experiments. Because HO-2 has some cysteine residues in its sequence, tsHO-2 homodimers (Mw: approximately 60 kDa) or homotrimers (Mw: approximately 90 kDa) were detected by non-reduced sodium dodecyl sulfate–polyacrylamide gel electrophoresis (SDS-PAGE). This may be related to the results of ultracentrifugation and size exclusion analyses [38]. Crosslinked heterodimers (Mw: approximately 100 kDa) were found in the non-reduced SDS-PAGE (Figure 1A) in the following combinations: T88C sCPR with tsHO-2 or V165C tsHO-2 or Q196C tsHO-2, and Q517C sCPR with Q196C tsHO-2 or Q196C/C264S/C281S tsHO-2 or Q196C/ΔHRM tsHO-2. As expected, Q517C sCPR formed a heterodimer with Q196C tsHO-2, whereas T88C sCPR formed a heterodimer with tsHO-2 and did not form a heterodimer with tsHO-2 containing ΔHRM or C264S/C281S mutation. Unexpectedly, the V165C mutation in tsHO-2 did not affect these results. This indicates that Gln517 of sCPR is in close proximity to Gln196 of tsHO-2, as expected, whereas Thr88 of sCPR is not in close proximity to Val165 of tsHO-2 but is in close proximity to Cys264 or Cys281 in HRM. We conclude that the FAD-binding domain of CPR, which includes Gln517, interacts with the catalytic domains of tsHO-2 and sHO-1, whereas HRM is spatially inserted between the catalytic domain of tsHO-2 and the FMN-binding domain of CPR, which includes Thr88 (Figure 1C). This is consistent with the previous NMR results of Zn protoporphyrin IX-bound human sHO-2, suggesting a redox-dependent interaction between the catalytic domain of sHO-2 and HRM [35]. Because tsHO-2 has a third cysteine residue (Cys126) which may be involved in the crosslinking with sCPR mutants, we carried out similar crosslinking experiments on the background of the C126S mutation (Appendix A). The results demonstrated that Cys126 of tsHO-2 is not involved in the crosslink with T88C or Q517C sCPR.

To evaluate whether Cys264 or Cys281 of tsHO-2 were in close proximity to Thr88 of the sCPR, a crosslinking experiment with C264S or C281S tsHO-2 was performed (Figure 2). Heterodimers were found at both C264S and C281S, indicating that both are in close proximity to Thr88 of CPR. Cys264 or Cys281 is presumed to bind to the second heme; therefore, we evaluated the quantity of heterodimers when the heme concentration varied. Comparison of lanes 9 and 10 or lanes 11 and 12 in Figure 2A shows that the addition of heme inhibited heterodimer formation. Cys264 and Cys281 were presumed to be the axial ligands of the heme iron of the second heme, thereby inhibiting heterodimer formation.

### 2.2. Enzymatic Assay

Crosslinking analysis demonstrated that the HRM of tsHO-2 was involved in its interaction with the sCPR. Therefore, the enzymatic activities of tsHO-2 and its HRM mutants in vitro were investigated to evaluate the effect of HRM (Figure 3A and Table 1). The enzymatic activity of tsHO-2 is approximately 40% relative to sHO-1 activity. HRM seemed to inhibit tsHO-2 activity because ΔHRM activity was comparable to sHO-1 activity. The disulfide bond between Cys264 and Cys281 would not be formed, and the second heme would bind to HRM in the assay because the purified tsHO-2 and its HRM mutants contained 5 mM tris (2-carboxyethyl) phosphine (TCEP) during purification, and an excess amount of the substrate heme was included in the assay conditions. When the tsHO-2 buffer was exchanged from the potassium phosphate buffer (pH 7.4), in which TCEP is unstable, to the buffer containing Tris-HCl (pH 7.4), potassium chloride, and TCEP to completely prevent the formation of the disulfide bond, tsHO-2 activity was also about 40% of ΔHRM tsHO-2 activity although the reason why tsHO-2 and ΔHRM tsHO-2 activity were decreased in this treatment was uncertain. This was consistent with the fact that C264S/C281S tsHO-2 activity was similar to ΔHRM and sHO-1 activity, C264S tsHO-2 activity was similar to tsHO-2 activity, and C281S tsHO-2 activity was between C264S/C281S tsHO-2 activity and C264S tsHO-2 activity. This demonstrated that heme-bound HRM inhibited sHO-2 activity. Previously, it was reported that HRM does not affect enzymatic activity based on the enzymatic assay of C264A, C281A, or its double mutants in rat HO-2 [39], which is inconsistent with our present results. This may be caused by the difference in redox conditions, the second heme binding to HRM, or the deletion of the N-terminal sequence in the present study.

### 2.3. Purification of Heme-Bound tsHO-2 Complex

Unlike sHO-1, sHO-2 can bind a second heme to HRM in addition to binding the substrate heme to the catalytic site. When tsHO-2 was prepared, TCEP was included in the purified enzyme to prevent the formation of disulfide bonds. To reconstitute heme-bound tsHO-2, five equimolar amounts of hemin were added so that heme could bind to both the catalytic site and the HRM. The UV-vis spectrum of the resultant heme–tsHO-2 complex showed a Soret peak at 410 nm and a shoulder at approximately 365 nm at pH 7.4 (Appendix A). Those of heme–ΔHRM tsHO-2 and heme–C264S/C281S tsHO-2 did not show the distinct shoulder around 365 nm; this shoulder suggests the heme binding to HRM as demonstrated in human HO-2 [32,33,34,35,38] and other HRM-containing proteins [40]. Compared with the heme–C281S tsHO-2 spectrum, the heme–C264S tsHO-2 spectrum showed a distinct shoulder at approximately 365 nm.

To obtain supporting data on heme binding to HRMs, the tsHO-2 and heme–tsHO-2 complex were individually digested with trypsin, and the fragmentation patterns of the HRM region were compared by mass spectrometry (Appendix A). The mass-to-charge ratio peak intensity at 2357, corresponding to the fragment containing both HRM1 and HRM2 (residues 263–284), was significantly reduced by complex formation with heme. Similarly, the peak at 2463, corresponding to the fragment containing His44, which is known as a proximal heme ligand in the catalytic site, was also significantly reduced. The peak intensities of fragments containing residues that were not expected to bind heme showed no significant changes in the absence or the presence of heme. These observations suggest that heme binding induces structural changes and specifically affects trypsin digestion in the regions involved in heme coordination.

### 2.4. Reduction Kinetics

As shown in the crosslinking analysis, HRM was spatially inserted between the catalytic domain of sHO-2 and the FMN-binding domain of sCPR in the sCPR–heme–tsHO-2 complex, implying that the distance between FMN and heme in the sCPR–heme–tsHO-2 complex was longer than that in the sCPR–heme–sHO-1 complex. This distance directly affected the reduction kinetics. To evaluate the reduction kinetics, the kinetics of formation of the CO-bound heme–tsHO-2 complex under CO saturated and anaerobic conditions were measured (Figure 3B and Table 1). Spectral changes upon CO binding are shown in Appendix A. The apparent reduction rate constant of heme–tsHO-2 was almost half of those of heme–ΔHRM tsHO-2 and heme–C264S/C281S tsHO-2, which were comparable to that of heme–sHO-1. The apparent reduction rate constant of heme–C264S tsHO-2 and heme–C281S tsHO-2 were in between that of heme–tsHO-2 and those of heme–ΔHRM tsHO-2 and heme–C264S/C281S tsHO-2. These results were consistent with those of the enzymatic assay.

### 2.5. Single-Turnover Analysis

The HO reaction was monitored spectroscopically. Following the reduction initiated in aerobic conditions, heme bound to HO is converted to a molecular oxygen-bound form which gives a typical Q-band absorption at 540 and 574 nm, then α-verdoheme (absorption at 680 nm) and CO-bound α-verdoheme (absorption at 638 nm) species appear, and finally, biliverdin (broad absorption around 700 nm) is formed. Typical examples of tsHO-2, ΔHRM tsHO-2, and sHO-1 are shown in Appendix A. During this reaction, the Soret band (absorption at approximately 400 nm) decreased drastically. The decay of the Soret band was a single exponential in sHO-1 and C264S tsHO-2, whereas it was a double exponential in tsHO-2 and its HRM mutants, except for C264S tsHO-2. A summary of the reaction kinetics is presented in Appendix A. The ratio of the fast and slow phases varied in the HRM mutants and tsHO-2. Fast phase was dominant in ΔHRM tsHO-2 and C264S/C281S tsHO-2, whereas slow phase was dominant in wild-type (tsHO-2), so that the single-turnover reaction catalyzed by tsHO-2 was apparently slower than those of sHO-1, ΔHRM tsHO-2, and C264S/C281S tsHO-2, which is consistent with the enzymatic assay and reduction kinetics results. Comparison of the kinetic constants implied that the decay of the Soret band of heme–sHO-1 was similar to the fast decay of heme–tsHO-2 and other HRM mutants, and that of heme–C264S tsHO-2 was similar to the slow decay of heme–tsHO-2 and other HRM mutants.

## 3. Discussion

We expected that the binding of CPR with HO-2 would be similar to that with HO-1 because the tertiary structures of the catalytic domains of heme–sHO-1 and heme–sHO-2 and the reaction mechanisms of HO-1 and HO-2 are similar. Previous NMR analyses were consistent with this expectation [29]. However, the present crosslinking analysis clearly demonstrates that the HRM of HO-2 is involved in its interaction with the CPR. This is inconsistent with the previous NMR analyses [29]; cross-peaks derived from residues located at the human HO-2 N and C termini (residues 1–28 and 243–288) were largely unaffected by the addition of equimolar CPR, although the NMR signals derived from Cys265-Pro266 and Cys282-Pro283, corresponding to Cys264-Pro265 and Cys281-Pro282 of rat HO-2, respectively, were unassigned. In that study, it was not clearly described whether the sample for NMR analysis bound the second heme. We believe that the sample for NMR analysis did not bind the second heme because the reductants to prevent disulfide bridge formation, such as dithiothreitol (DTT) or TCEP, were not present during purification [29], so the results are inconsistent. MALDI-TOF MS analysis of our samples demonstrated that the HRM region was protected from trypsin digestion by heme binding, suggesting a second heme binding in the HRM region.

The conformation of HRM is expected to vary depending on the cellular redox conditions and the quantity of free heme; therefore, HRM is expected to regulate the interaction between HO-2 and the CPR. It is difficult to prepare sHO-2 where heme is bound to the catalytic pocket only; however, it is considered that ΔHRM or C264S/C281S tsHO-2 mimics tsHO-2 where heme is bound to the catalytic pocket only with no disulfide bond between Cys264 and Cys281. The present enzymatic analysis clearly indicated that heme-bound HRM negatively regulates HO-2 activity in vitro. Crosslinking analysis suggested that HRM is located between the FMN-binding domain of CPR and the catalytic domain of HO-2, implying that the distance between FMN and heme bound to the catalytic domain of sHO-2 is slightly longer than the distance between FMN and that of sHO-1. The redox kinetics results were consistent with this expectation; the reduction in heme–tsHO-2 was slower than that in the heme-bound HRM mutants and heme–sHO-1. The results of the single-turnover analysis of tsHO-2 and its HRM mutants were consistent with those of the enzymatic analysis. Although the reason why the decay of Soret bands is biphasic in heme–tsHO-2 and monophasic in heme–sHO-1 is uncertain, it may be caused by a subtle difference in the kinetics of the sHO-1 and tsHO-2 reactions or by the transfer of heme from the HRM site to the catalytic site [32]. It is well known that inducible HO-1 activity is regulated by several factors via transcription factors such as Keap1-Nrf2, Bach1, and HIF1α; however, the regulatory factor of constitutive HO-2 is uncertain. The present results suggest that HO-2 activity is regulated by cellular redox conditions and free heme concentration through conformational changes in the HRM, which regulate its interaction with the CPR.

## 4. Materials and Methods

### 4.1. Protein Expression and Purification

HO-1, HO-2, and CPR are membrane-bound proteins anchored to the cytoplasmic surface of the endoplasmic reticulum. To handle these conveniently, we prepared soluble forms of rat HO-1, HO-2, CPR (sHO-1, sHO-2, and sCPR), and ΔTGEE by removing their membrane-spanning regions. Rat sHO-1, sCPR, ΔTGEE, and biliverdin reductase were expressed in *Escherichia coli* (*E. coli*) and purified as described earlier [24,25,41,42,43,44]. An open reading frame (ORF) sequence of the previously constructed pET-11b-sHO-2 (Met1-Lys292) plasmid [45] was cut out using NdeI and BamHI restriction enzymes (New England Biolabs, Ipswich, MA, USA), and the resultant ORF of sHO-2 was subcloned into a pET-15b vector for the expression of the 6xHis-tagged enzyme (Merck, Darmstadt, Germany), then the DNA sequence corresponding to Met1-Lys28 was deleted using appropriate primers (29-293-f and 29-293-r) and KOD -Plus- mutagenesis kit (Toyobo, Osaka, Japan) (Appendix A). Met1-Lys28 is disordered in the crystal structure [19]. NMR analysis [35] of heme–human HO-2 (Appendix A) and a splicing variant missing this region (P30519-2, UniProt) have been reported in humans. The resultant plasmids were used for templates for site-directed mutagenesis to produce expression vectors for a series of HRM mutants (ΔHRM, C126S/ΔHRM, C264S, C281S, C264S/C281S, and C126S/C264S/C281S enzymes) using appropriate primers and KOD -Plus- mutagenesis kit (Appendix A and Appendix A). The ORF in the pET-15b-tsHO-2 or pET-15b-HRM mutants was excised using NdeI and BamHI, and the resultant ORF of tsHO-2 or its HRM mutants was subcloned into a pET-21a(+) vector (Merck) to express the non-His-tagged enzyme. The pET-15b-tsHO-2, pET-21a-tsHO-2, pET-15b-HRM mutants, and pET-21a-HRM mutants plasmids, in which the ORF sequences were confirmed by DNA sequencing, were transformed into *E. coli* C41 (DE3) (Merck), followed by overnight culturing of the transformed *E. coli* at 37 °C on Luria–Bertani (LB) agar plates containing 100 μg/mL ampicillin. A single colony was selected to inoculate 2 mL of LB medium containing 100 μg/mL ampicillin. The cells were cultured in Terrific Broth medium containing 100 μg/mL ampicillin at 0.4% *v/v* inoculum using baffled flasks with rigorous shaking at 30 °C for 3 h, then isopropyl β-D-1-thiogalactopyranoside (IPTG) was added to a final concentration of 100 μM to initiate tsHO-2 expression. The harvested cells from the overnight culture were stored at −80 °C until use.

Cells expressing 6xHis-tagged tsHO-2 and its HRM mutants were disrupted using BugBuster protein extraction reagent (Merck) supplemented with a protease inhibitor cocktail (EDTA free) (Nacalai Tesque, Kyoto, Japan). The insoluble fraction was removed by centrifugation, and the soluble fraction was immediately purified using Ni-NTA agarose (His-Accept, Nacalai Tesque). After washing the Ni-NTA agarose with a buffer containing 50 mM Tris (pH 7.4) and 150 mM KCl, the bound proteins were eluted with a buffer supplemented with 500 mM imidazole-HCl (pH 8.0). Excess imidazole was removed by ultrafiltration, and the resultant purified proteins were immediately used for crosslinking experiments to prevent the formation of the disulfide bond between Cys264 and Cys281 and to limit proteolysis of the C-terminal sequence of tsHO-2 and its HRM mutants.

Cells expressing non-His-tagged tsHO-2 and its HRM mutants were sonicated in a solution containing 50 mM Tris-HCl (pH 8.0), 2 mM ethylenediaminetetraacetic acid (EDTA) (pH 8.0), 1 mM DTT, and a protease inhibitor cocktail (EDTA free) (Nacalai Tesque). The insoluble fraction was then removed by centrifugation. tsHO-2 and its HRM mutants were purified using the method used for sHO-1 purification [42], using ammonium sulfate fractionation and anion exchange, size exclusion, and hydroxyapatite columns. Briefly, the supernatant was fractionated with 30–60% ammonium sulfate, and the insoluble pellet was solubilized with the buffer containing 20 mM Tris-HCl (pH 7.4), 0.2 mM EDTA, and 1 mM DTT. The protein solution was dialyzed against the same buffer and loaded onto a Hitrap Q HP column (Cytiva, Tokyo, Japan) equilibrated with the same buffer. The blue-greenish fractions, to which biliverdin was bound, were eluted using a linear gradient of potassium chloride and collected. Subsequently, the collected fractions were concentrated and introduced onto a Sephacryl S-200 HR column (Cytiva) equilibrated with 20 mM potassium phosphate (pH 7.4) and 5 mM TCEP-NaOH (pH 7.0). Excess biliverdin was partially removed using a Bio-Scale Mini CHT Type I Cartridge (Bio-Rad, Hercules, CA, USA) equilibrated with 20 mM potassium phosphate (pH 7.4) and 5 mM TCEP-NaOH (pH 7.0). The resulting purified proteins were eluted with a linear gradient of potassium phosphate and collected for enzymatic assays and reconstitution of heme-bound proteins. In addition to DNA sequencing of the ORF, C264S, C281S, and C264S/C281S mutations were confirmed by MALDI-TOF MS analysis of trypsin-digested fragments of recombinant proteins (Appendix A).

The heme–tsHO-2 complex and heme–HRM mutants were reconstituted with five equimolar amounts of heme, and the buffer was exchanged by ultrafiltration and purified by chromatography on a hydroxyapatite column (Bio-Rad), similar to heme–sHO-1 [46]. The concentrations of heme–tsHO-2 and heme–HRM mutants were determined using a BCA protein assay kit (Nacalai Tesque) following the manufacturer’s protocol.

### 4.2. Crosslinking Experiments

Previously, we demonstrated that Val146 and Lys177 of sHO-1 are adjacent to Thr88 and Gln517 of sCPR in the ΔTGEE–heme–sHO-1 complex structure, and crosslinked heterodimers were identified when those were substituted by cysteine [25]. To elucidate whether the interaction between sCPR and tsHO-2 is similar to that between sCPR and sHO-1, we substituted Val165 or Gln196 of tsHO-2 with cysteine, which corresponds to Val146 or Lys177 of sHO-1, respectively. The 6xHis-tagged V165C or Q196C tsHO-2 and their HRM mutants were prepared as described above. The 6xHis-tagged T88C or Q517C sCPR were prepared as described previously [25]. tsHO-2 or HRM mutants were mixed with 1.5 equimolar of hemin, 0.5 mM NADP^+^, and T88C or Q517C sCPR overnight at 4 °C unless otherwise stated. The mixtures were analyzed by SDS-PAGE under non-reduced and reduced conditions. The resulting gels were stained with Coomassie Brilliant Blue R-250.

### 4.3. Enzymatic Assay

HO activity was determined based on the rate of bilirubin formation, which was monitored by the increase in the absorption at 468 nm at 37 °C [47]. Bilirubin formation from biliverdin is catalyzed by biliverdin reductase present in the assay mixture. Assay mixture contained 40 μM heme, 0.125-0.5 μM sHO-1 or tsHO-2 or its HRM mutants, 2.9 μM sCPR, 0.5 mg/mL bovine serum albumin (Merck), 165 μg/mL bovine catalase (Fujifilm Wako Pure chemicals Corp., Osaka, Japan), and 5.3 μg/mL rat biliverdin reductase, in 0.1 M potassium phosphate buffer (pH 7.4). The assay was initiated by the addition of 100 µM NADPH.

Single-turnover reactions were monitored based on changes in the absorption spectra at 30 °C. The reaction mixtures (0.1 mL) consisted of 4.3 µM heme–sHO-1 or heme–tsHO-2 or heme-bound HRM mutants, 40 nM sCPR, and 100 µM NADPH in 0.1 M potassium phosphate buffer (pH 7.4). The spectra were recorded over a range of 300–900 nm at 2 min intervals. Exponential fitting of Soret decay was performed using GraphPad Prism software version 7.04.

Heme reduction was monitored based on changes in the absorption spectra at 30 °C in a CO-saturated anaerobic atmosphere. The CO-saturated reaction mixtures (0.1 mL) were composed of 4.3 µM heme–tsHO-2, or heme-bound HRM mutants, 1–4 nM sCPR, and 100 µM NADPH in 0.1 M potassium phosphate buffer (pH 7.4). The reaction was initiated by the addition of NADPH. The initial rate of ferric heme reduction was calculated based on the increase in absorbance at 418 nm. The differences in molar absorption coefficient (ε) between the ferric heme–tsHO-2 and CO-bound ferrous heme–tsHO-2 were 60.6 mM^−1^ cm^−1^ at 418 nm. That of ΔHRM, C264S, C281S, and C264S/C281S were 105.9, 48.9, 72.3, and 94.8 mM^−1^ cm^−1^, respectively. The concentration of NADPH used in all assays was determined by measuring the absorbance at 340 nm (ε = 6.22 mM^−1^ cm^−1^). A UV-2550 spectrophotometer (Shimadzu, Kyoto, Japan) was used for spectroscopic measurements of the enzymatic assays.

### 4.4. MALDI-TOF MS Analysis

Fragmentation of tsHO-2 and its mutants for the MALDI-TOF MS analysis was carried out with Trypsin Gold, Mass spectrometry Grade (Promega, Madison, WI, USA) in 20% acetonitrile and 70 mM ammonium bicarbonate at 37 °C for 7 h. The digests were concentrated and desalted using a Zip-Tip C18 (Merck). 2, 5-dihydroxybenzoic acid (Fujifilm Wako Pure Chemicals Corp.) was used as a matrix. Mass spectra were acquired using an Autoflex maX (Bruker Daltonics, Bremen, Germany) controlled by FlexControl software version 3.4 and analyzed using FlexAnalysis software version 3.4.

## Figures and Tables

**Figure 1 ijms-26-02318-f001:**
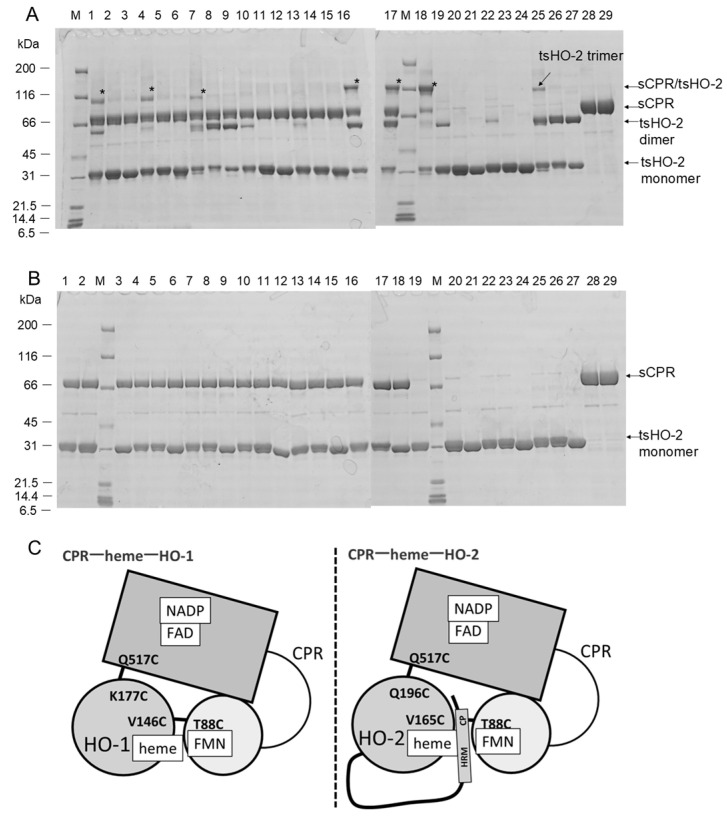
Crosslinking analysis of 6xHis-tagged rat tsHO-2 and rat sCPR. (**A**) SDS-PAGE in non-reduced condition. Asterisks indicate the bands of sCPR/6xHis-tagged tsHO-2 heterodimer. Implications of each band are shown on the right side of the gels. Applied samples in each lane are shown below; sCPR mutant and tsHO-2 (or its mutant) are applied in lanes 1–18, and tsHO-2 (or its mutant) only are applied in lanes 19–27. sCPR of lanes 1–9 and lanes 10–18 are T88C sCPR and Q517C sCPR, respectively. tsHO-2 applied in lanes 1–9 is shown below. 1: tsHO-2 (wild-type), 2: C264S/C281S, 3: ΔHRM, 4: V165C, 5: V165C/C264S/C281S, 6: V165C/ΔHRM, 7: Q196C, 8: Q196C/C264S/C281S, and 9: Q196C/ΔHRM. tsHO-2 of lanes 10–18 and lanes 19–27 are the same as tsHO-2 of lanes 1–9. T88C sCPR and Q517C sCPR without tsHO-2 are applied in lanes 28 and 29, respectively. Lane M shows the protein markers for SDS-PAGE (Nacalai Tesque). (**B**) SDS-PAGE in reduced condition. The samples applied to each lane were the same as indicated for Figure 1A, but with 2-mercaptoethanol. (**C**) Schematic diagrams of the interaction of rat CPR and rat HO-1 or rat HO-2. The rectangle box and light-gray circle show the FAD-binding domain and FMN-binding domain, respectively. The gray circle shows the catalytic domain of HO-1 or HO-2. CPR–heme–HO-1 model reflects the crystal structure of ΔTGEE–heme–HO-1 complex and crosslinking analysis [25]. Our studies with cysteine-substituted mutants demonstrated that Gln196 of HO-2 is located in proximity of Gln517 of CPR. Bold lines between CPR and HO are disulfide bonds between cysteine-substituted mutants of CPR and HO.

**Figure 2 ijms-26-02318-f002:**
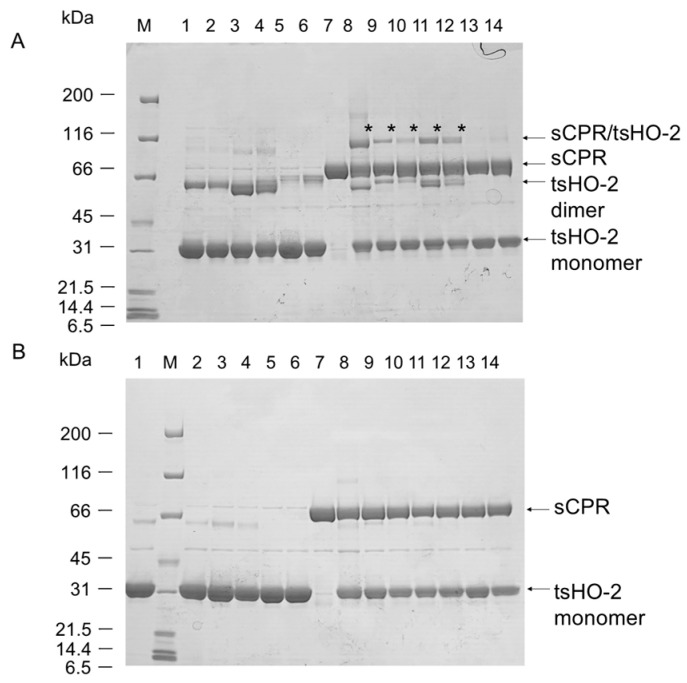
Crosslinking analysis of 6xHis-tagged C264S or C281S tsHO-2 mutants with T88C sCPR mutant. (**A**) SDS-PAGE in non-reduced condition as shown in Figure 1A. Applied samples in each lane are shown below; T88C sCPR and tsHO-2 (mutants) are applied in lanes 8–14, tsHO-2 (mutants) only are applied in lanes 1–6, and T88C sCPR only is applied in lane 7. The tsHO-2 mutants applied in lanes 1–6 are shown below. 1 and 2: C264S, 3 and 4: C281S, and 5 and 6: C264S/C281S. Hemin is included in 5 equimolar of tsHO-2 mutants in even numbers of lanes 1–6 and 9–14. tsHO-2 (wild-type) was applied in lane 8. tsHO-2 applied in lanes 9–14 is the same as lanes 1–6. Lane M shows the protein markers. (**B**) SDS-PAGE in reduced condition. The applied sample in each lane is the same as Figure 2A, but with 2-mercaptoethanol.

**Figure 3 ijms-26-02318-f003:**
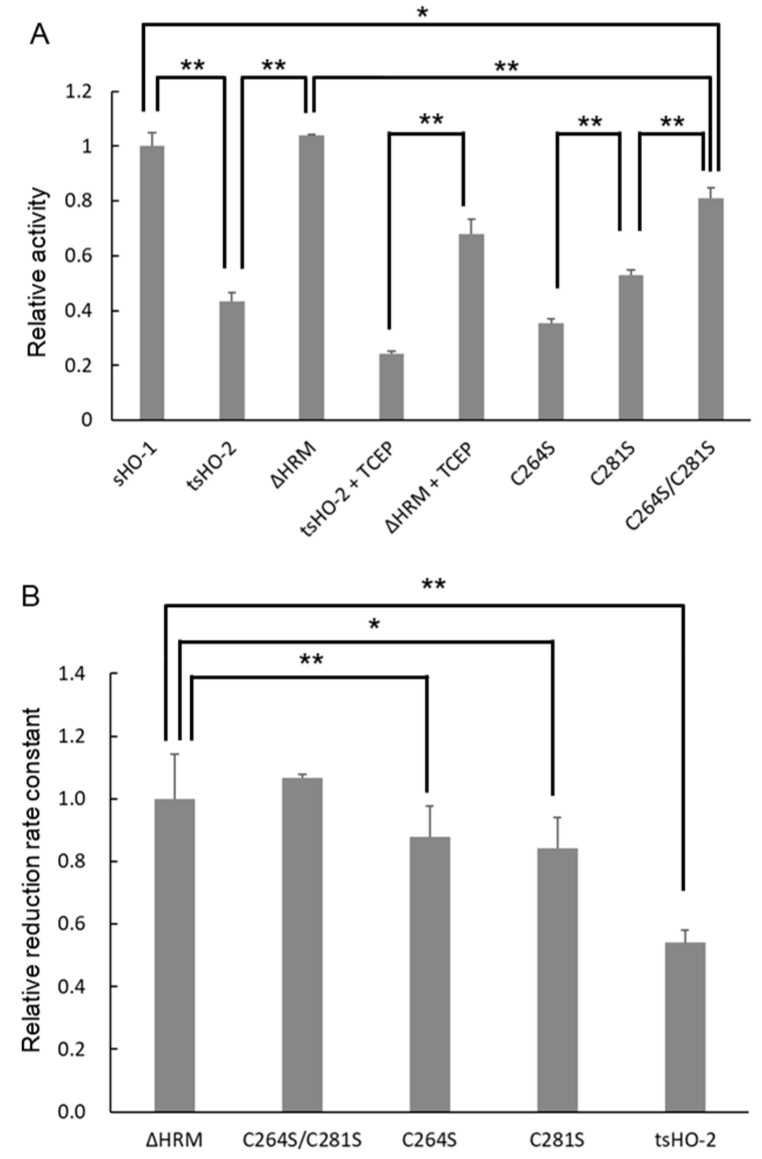
Enzymatic assay and reduction kinetics of non-His-tagged tsHO-2 and its HRM mutants. (**A**) The vertical axis indicates the enzymatic activity relative to that of sHO-1. In the + TCEP conditions, the buffer of tsHO-2 or ΔHRM was exchanged to the buffer containing 20 mM Tris-HCl (pH 7.4), 150 mM KCl, and 5 mM TCEP-NaOH. TCEP-contained tsHO-2 was stored overnight in the anaerobic chamber and was immediately applied for enzymatic assay. The final assay condition contains approximately 1% tsHO-2 buffer containing Tris, KCl, and TCEP. Error bars represent standard errors (N = 3). Single and double asterisks indicate that the *p*-values between the data are less than 0.05 and 0.01, respectively. (**B**) The vertical axis indicates the reduction kinetics relative to that of ΔHRM tsHO-2. Error bars represent standard errors (N = 4).

**Table 1 ijms-26-02318-t001:** Summary of the enzymatic assays and reduction kinetics.

	Enzymatic Activity			Apparent Reduction Rate Constant	
	(min^−1^)	Relative to sHO-1 (%)		(min^−1^)	Relative to ΔHRM (%)
sHO-1	13.8 ± 0.69	100 ± 4.99		122 ± 3.8 [27]	115 ± 3.1
tsHO-2	6.01 ± 0.43	43.5 ± 3.09		57.5 ± 2.30	54.1 ± 4.0
ΔHRM	14.3 ± 0.069	104 ± 0.50		106 ± 15.3	100 ± 14.4
C264S	4.91 ± 0.22	35.5 ± 1.57		93.2 ± 9.24	87.7 ± 9.9
C281S	7.32 ± 0.27	53.0 ± 1.96		89.6 ± 8.81	84.3 ± 9.8
C264S/C281S	11.2 ± 0.56	80.8 ± 4.08		113.2 ± 1.40	107 ± 1.2

## Data Availability

The data presented in this study are available upon request from the corresponding author.

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
