# Peer review of "Heme Regulatory Motif of Heme Oxygenase-2 Is Involved in the Interaction with NADPH–Cytochrome P450 Reductase and Regulates Enzymatic Activity"

_ijms, 2025, doi:10.3390/ijms26052318_

Round 1

Reviewer 1 Report

Comments and Suggestions for Authors

This manuscript presents a well-designed and exemplary performed study of interactions between HMOX2 and CPR. I have only two concerns which I would like the authors to address:

  1. Rat HMOX2 contains three cysteine residues - C126, C264, and C281. According to the AlphaFold structural model, all three residues are surface-exposed and, therefore, may be involved in intermolecular S-S-bonding. However, in their analysis of S-S-bonding crosslinks, the authors completely ignore the presence of Cys-126. Why? The authors should mention Cys-126 and explain the reason for ignoring it in analyzing intermolecular S-S-bonding.  
  2. The explanation for the lack of activating effect of TCEP on the activity of ts-HO2 (lines 178-179) does not sound convincing. The authors suggested that TCEP itself may inhibit the enzyme. However, it is unclear what may be the mechanism of this inhibition.  To test their hypothetic explanation, the authors may perform a simple experiment: they could probe the effect of TCEP on the activity of C264S/C281S mutant (or, better, on the activity of C126S/C264S/C281S triple mutant). If their explanation is right, TCEP will inhibit the activity in this case.  The addition of this simple control will make the analysis of the effect of C264S/C281S more convincing.
Comments on the Quality of English Language

Although the manuscript is written in good English and reads easily, I found several minor issues that may need correction:

  1. Lines 115-116: replace “in the combination shown below;” with “in the following combinations:”
  2. Lines 140-141: replace “Applied sample in each lane is same as Figure 1A” with “The samples applied to each lane were the same as indicated for Figure 1A”
  3. Line 144: Remove “is” in “Cysteine-substituted mutants showed Gln196 of HO-2 is located nearby Gln517” – it should be “Cysteine-substituted mutants showed Gln196 of HO-2 located nearby Gln517”. However, a better version would be: “Our studies with cysteine-substituted mutants demonstrated that Gln196 of HO-2 is located in a proximity of Gln517”.
  4. Insert dashes in “molecular oxygen bound” (line 234): “molecular-oxygen-bound”
  5. Legend to Figure S4 (Supplementary material): Replace “UV-vis spectrum change” with “UV-VIS spectral changes”. Also, replace “although the concentration of CPR (40 nM) is same” with “although the concentration of CPR (40 nM) was the same.”

Reviewer 2 Report

Comments and Suggestions for Authors

The authors performed crosslinking experiments on the complex of a constitutively expressed isoform of mammalian heme oxygenase (HO-2) with NADPH-cytochrome P450 reductase (CPR), proposing that heme regulatory motif (HRM) is located near the FMN-binding domain of CPR in the complex. In addition, enzymatic assay, and reduction kinetics suggest that heme-bound HRM negatively regulates HO-2 activity in vitro, indicating the regulation of HO-2 activity by cellular redox conditions and free heme concentrations. The experimental methods used in the manuscript are reasonable, and the conclusion is supported by the experimental data. This manuscript is suitable for publication in International Journal of Molecular Science. However, I would like the authors to consider the following issues before publication.

  1. The authors mentioned that the experimental results in the present study are inconsistent with the previous NMR analyses (Ref.29) because the sample of NMR analysis did not bind the second heme due to the difference in the experimental condition such as the absence of the use of reductants to prevent disulfide bridge formation like dithiothereitol. It's not a strong request, but would it be hard to present experimental data on the second heme binding under the experimental conditions in Ref. 29?

  1. If possible, could you show a second heme-bound tertiary structure predicted by AlphaFold3? (It's not mandatory.)
